# Usefulness of Implementation Outcome Scales for Digital Mental Health (iOSDMH): Experiences from Six Randomized Controlled Trials

**DOI:** 10.3390/ijerph192315792

**Published:** 2022-11-27

**Authors:** Erika Obikane, Natsu Sasaki, Kotaro Imamura, Kyosuke Nozawa, Rajesh Vedanthan, Pim Cuijpers, Taichi Shimazu, Masamitsu Kamada, Norito Kawakami, Daisuke Nishi

**Affiliations:** 1Department of Mental Health, Graduate School of Medicine, The University of Tokyo, Tokyo 113-0033, Japan; 2Department of Social Medicine, National Center for Child Health and Development, Tokyo 157-0074, Japan; 3Department of Digital Mental Health, Graduate School of Medicine, The University of Tokyo, Tokyo 113-8655, Japan; 4Department of Psychiatric Nursing, Graduate School of Medicine, The University of Tokyo, Tokyo 113-0033, Japan; 5Department of Population Health, Grossman School of Medicine, New York University, New York, NY 10016, USA; 6Department of Clinical, Neuro and Developmental Psychology, Vrije Universiteit Amsterdam, 1081 Amsterdam, The Netherlands; 7Division of Behavioral Sciences, Institute for Cancer Control, National Cancer Center, Tokyo 04-0045, Japan; 8Department of Health Education and Health Sociology, Graduate School of Medicine, The University of Tokyo, Tokyo 113-0033, Japan

**Keywords:** implementation outcomes, acceptability, appropriateness, feasibility, satisfaction

## Abstract

Objectives: Measuring implementation outcomes for digital mental health interventions is essential for examining the effective delivery of these interventions. The “Implementation Outcome Scale of Digital Mental Health” (iOSDMH) has been validated and used in several trials. This study aimed to compare the iOSDMH for participants in six randomized controlled trials (RCTs) involving web-based interventions and to discuss the implications of the iOSDMH for improving the interventions. Additionally, this study examined the associations between iOSDMH scores and program completion rate (adherence). Methods: Variations in total scores and subscales of the iOSDMH were compared in six RCTs of digital mental health interventions conducted in Japan. The web-based intervention programs were based on cognitive behavioral therapy (2 programs), behavioral activation (1 program), acceptance and commitment (1 program), a combination of mindfulness, behavioral activation, and physical activity (1 program), and government guidelines for suicide prevention (1 program). Participants were full-time employees (2 programs), perinatal women (2 programs), working mothers with children (1 program), and students (1 program). The total score and subscale scores were tested using analysis of variance for between-group differences. Results: Total score and subscale scores of the iOSDMH among six trials showed a significant group difference, reflecting users’ perceptions of how each program was implemented, including aspects such as acceptability, appropriateness, feasibility, overall satisfaction, and harm. Subscale scores showed positive associations with completion rate, especially in terms of acceptability and satisfaction (R-squared = 0.93 and 0.89, respectively). Conclusions: The iOSDMH may be a useful tool for evaluating participants’ perceptions of features implemented in web-based interventions, which could contribute to improvements and further development of the intervention.

## 1. Introduction

Digital mental health interventions have rapidly become available worldwide, with recent studies [1,2,3,4] finding they effectively prevent and improve various mental health outcomes. Although digital mental health interventions can be a key solution to the shortage of mental health care providers and the stigma of medical visits for mental health problems, these interventions face challenges related to insufficient implementation, including low program adherence and high attrition rates.

We previously developed and validated the implementation Outcome Scales of Digital Mental Health (iOSDMH) for users (i.e., people who use the program or patients) to evaluate implementation aspects of mental health interventions delivered via digital and telecommunication technologies such as internet websites, movies, apps, and e-mails [5]. Although different measurement tools exist (e.g., system usability scale [6]), there was few scales for measuring implementation outcomes comprehensively, focusing on digital mental health. We thus developed iOSDMH based upon Proctor’s implementation conceptual frameworks [7,8], which reflects exist literature comprehensively, and related research [9,10,11,12] in order to assess indicators of implementation success, implementation processes, and intermediate outcomes linked to effectiveness or quality outcomes.

The iOSDMH for users includes 19 implementation items, including acceptability, appropriateness, feasibility, satisfaction, and harm. However, it remains unclear which implementation items are more predictive of completion rate or participant attitude compared with other items. A previous literature review indicated a positive association between treatment satisfaction and adherence, compliance, or persistence [13]. This association was partially explained by the mechanism of the theory of reasoned action (TRA) developed by Martin Fishbein and Icek Ajzen (1975, 1980), in which a positive preexisting attitude and subjective norms promote behavioral intentions [14]. Therefore, ‘users’ beliefs and values about the impact of receiving an intervention, which influences the evaluation of satisfaction, might be important in achieving high completion in digital mental health. Although satisfaction and other implementation outcomes can be influenced by many aspects of an intervention (e.g., efficacy, side effects, communication with health care providers, personal treatment history), these outcomes might increase intervention effectiveness if improving implementation outcomes indeed increases program completion rates. However, to our knowledge, there is no available evidence showing an association between implementation outcome and completion rate in digital mental health interventions. Nor is it known how to utilize profile patterns of the iOSDMH to improve the completion rate of future web-based interventions. We administered the iOSDMH to intervention group participants in six randomized controlled trials (RCTs) of digital mental health interventions conducted in Japan that included the iOSDMH. Our aim was two-fold: (1) to investigate the usefulness of iOSDMH total scores or subscale scores in differentiating implementation aspects of each intervention; and (2) to determine their association with completion rates. We further discussed the interpretation and variations of iOSDMH scores and how such findings can improve program contents or intervention methods for future investigation.

## 2. Methods

### 2.1. Study Design

We compared variations of implementation outcomes in six RCTs of digital mental health interventions (with their scales). Table 1 presents the study characteristics and completion rates of the RCTs. These six trials were registered and/or the protocols were published elsewhere [15,16,17,18,19,20]. Table 1 also presents the time points at which the iOSDMH was measured in each study. All study procedures were approved by the Research Ethics Review Board of the Graduate School of Medicine, University of Tokyo, and each study utilized the same scale (iOSDMH) to measure implementation outcomes.

Study 1 was an app-based self-help Cognitive Behavioral Therapy (CBT) intervention for pregnant women [18]. Study 2 was a web-based self-help acceptance and commitment therapy (ACT) intervention with a writing exercise for working mothers with a small child [20]. Study 3 was a machine-guided self-help CBT intervention with a writing exercise for workers [17]. Study 4 was a psychoeducational intervention using a website for workers [16]. Study 5 was a video-based gatekeeper program for students to prevent suicide of their peers. Study 6 was a web-based behavioral activation therapy (BA) intervention with a writing exercise for postnatal women [19].

### 2.2. Measurement Variables

#### 2.2.1. Implementation Outcome Scales for Digital Mental Health (iOSDMH)

The iOSDMH has several distinct versions for users, providers, and managers. This study utilized the users’ version, which comprises two parts: (1) evaluations (14 items) and (2) adverse events of using digital mental health programs (5 items). Each item’s response was scored on a 4-point Likert-type scale ranging from 1 (disagree) to 4 (agree). The subscales interpreted “relatively agree” and “agree” as being implemented (preferable). Evaluations with their number of items and possible score ranges were as follows: acceptability (3 items; 3–12), appropriateness (4 items; 4–16), feasibility (6 items; 6–24), harm (5 items; 5–20), and satisfaction (1 item; 1–4). The total score has 14 items (14–56, excepting the harm items). Scores were calculated by summing the items’ scores. The original development paper [5] calculated the total score by summing all 19 items, which we changed so that a high score of 14 items signified good implementation and 5 harm items signified less favorable implementation. Item 9 was reversed before summing. Inclusion of reversed scale items enhances scale validity because it strategically drives respondents to attend more carefully to specific content of individual items [21].

#### 2.2.2. Details of the Intervention Studies

The study characteristics were collected as descriptive data: research design (target population, total number of study participants, recruitment method, primary aim of the intervention, primary outcome), intervention details (intervention type, basal theory of intervention, number of sessions, learning time per session, intervention duration, content type, homework/exercises, availability, interactions with professionals/other participants), facilitations and functions (timing of new module reminder, additional reminders for non-learners, participants’ reward, reward for questionnaire completion), and findings and presentations (program completion rate, respondents’ characteristics for the iOSDMH, timing of iOSDMH measurement).

### 2.3. Statistical Analysis

We determined whether iOSDMH scores can differentiate implementation outcomes among the studies by testing group differences with a chi-square test for the proportion implemented in each item. We tested total score and subscale scores using analysis of variance for group differences. The association between implementation subscales and completion rate was assessed by calculating the R-squared value using Microsoft Excel (Microsoft, Redmond, WA, USA). The statistical significance for all analyses in this study was set at 0.05 (two-tailed), and 95% confidence intervals were calculated. All statistical analyses were performed using the Japanese version of SPSS 28.0 (IBM Corp., Armonk, NY, USA).

## 3. Results

Table 2 shows the descriptive scores and response rates indicating users’ positive evaluations. All items demonstrated significant group differences (item 15, *p* = 0.029; others, *p* < 0.001).

For acceptability, Study 5 demonstrated the highest score for all items among the six trials. Study 5 was an online student peer gatekeeper program that provided basic knowledge about suicide prevention via YouTube video for students willing to become peer supporters of suicide prevention. Study 3 showed the second highest scores for all items in acceptability. Studies 1, 2, and 6 were e-learning programs targeting a specific population. These studies presented similar profile patterns for three items in acceptability: (1) positive evaluation rates fell between 68.9% and 79.7% for item 1 (*advantages outweigh the disadvantages for keeping my mental health*) and item 3 (*acceptable for me*); and (2) low evaluation rates (positive evaluation rates between 30.0% and 51.8%) for item 3 (*improves my social image*). Study 4 was an e-learning program for full-time workers that encouraged participants to read webpages of interest. This study presented intermediate scores (positive evaluation rates between 67.2% and 77.2%) for items 1 (*advantages outweigh the disadvantages for keeping my mental health*) and 3 (*acceptable for me*) and a low evaluation score (44.7% positive evaluation rate) for item 2 (*improves my social image*).

For appropriateness, Study 5 demonstrated the highest evaluation scores for three of four items among all the programs: item 4 (*appropriate [from your perspective, it is the right thing to do]*), item 6 (*suitable for my social condition),* and item 7 (*fits my living condition*). The studies on prenatal (Study 1) and postnatal (Study 6) experiences demonstrated a positive evaluation rate above 80% in appropriateness-related items, especially for item 5 (*applicable to my health status*). Study 2 presented high user evaluations (77.6%) for item 4 (*appropriate [from your perspective, it is the right thing to do]*) and moderate user evaluations for other items. Study 3 also presented high evaluations (84.6%) for item 4 (*appropriate [from your perspective, it is the right thing to do]*) and moderate evaluations for other items. Study 4 demonstrated a low evaluation for item 5 (*applicable to my health status*) and item 7 (*fits my living condition*), with a positive evaluation rate of 50.2% and 48.9%, respectively.

For feasibility, Study 1, Study 3, and Study 5 demonstrated higher user evaluations for all items compared with other programs. Study 2 reported moderate to high feasibility (positive evaluation rates between 60.2% and 77.2%) except for item 9 (*physical effort*; 32.0%). Study 4 showed moderate user evaluations (positive evaluation rates between 57.4% and 67.1%). Study 6, which required the longest time per session and the longest total time to complete the whole program, reported low scores for item 8 (*easy to use*), item 9 (*physical effort)*, item 10 (*total length is implementable*), item 11 (*length of one content is implementable*), and item 13 (*easy to understand*), with positive evaluation rates of 58.6%, 34.5%, 41.3%, 48.3%, and 41.4%, respectively.

As for overall satisfaction, Study 5 reported the highest scores. For harms, Studies 2, 4, 5, and 6 reported high rates of concern that the program was time-consuming (over 25%), while Studies 2 and 6 reported that users regularly perceived excessive learning-related stress (negative evaluation rates of 52.5% and 41.4%, respectively).

Table 3 shows iOSDMH total scores and subscale scores, which were characterized by significant group differences. Study 5 presented the highest total score as well as the highest subscale scores in acceptability, appropriateness, feasibility, and satisfaction. Study 1 and Study 3 presented the second highest total scores. 

Figure 1 shows the association between iOSDMH subscale scores and total iOSDMH scores (excluding harm scores) with completion rates. Subscale scores and completion rates showed a nearly linear trend. Acceptability and satisfaction were highly associated with completion rate (*R*-squared value = 0.93, *R*^2^ = 0.89; respectively). Harm showed a weak inverse association. iOSDMH total scores also showed a high association with completion rate (*R*-squared value = 0.95).

## 4. Discussion

This study evaluated implementation and dissemination aspects of six RCTs of digital mental health programs using iOSDMH scales as developed in a published study. The iOSDMH was shown to be an effective tool for understanding program characteristics on acceptability, appropriateness, feasibility, overall satisfaction, and harm as well as user evaluations of these points. Moreover, the iOSDMH was found to effectively measure prediction of program completion rates with moderate to high associations. Compared with appropriateness and feasibility, acceptability and satisfaction were more strongly associated with completion rate. These findings might indicate that subjective positive feelings about a program are important for adherence. Examination of iOSDMH total scores or subscales made it possible to clarify future assignments to promote further program implementation.

### 4.1. Acceptability

Acceptability considers whether a program’s users feel that the program benefits their mental health and they have a positive impression of the program.

Acceptability relates to various factors, including a program’s topics, theory, and participation style. Study 5 (targeting students interested in suicide prevention) seemed to attract users’ interest and utilized effective delivery methods such as peer role play and YouTube, resulting in high acceptability scores. YouTube videos and peer role play might have contributed to these high evaluations of acceptability, a finding that is consistent with a previous study [22]. Study 3 (e-learning programs with AI feedback customized to users) received the second highest acceptability scores. Individualized program support might have led to high program acceptance. Moreover, the programs that focused on concerns raised by specific populations of women (Studies 1, 2, and 6) also reported high user acceptability. Topic selection that matches users’ characteristics is essential for increasing user acceptability. In contrast, Study 4 resulted in intermediate acceptability even though this study adopted a unique strategy that allowed users to tailor the program’s order according to their interests. However, the topics covered might have been too general and not aimed at a specific population.

Our study showed high associations between acceptability and completion rate, which is consistent with previous studies showing that meeting client needs and preferences, sharing decision-making, and tailoring care are important for improving implementation and effectiveness [23,24]. Our study suggests directions for future research in that personalized or tailored programs can lead to increased acceptability to maximize completion rates.

### 4.2. Appropriateness

We found that matching program content with a user’s physical or medical needs is an important factor for appropriateness. In Study 5, students willing to be gatekeepers for suicide prevention might have needed communication skills for suicide prevention and thus felt that the program content was appropriate and suited their social condition. They might have perceived that communication skills useful for listening to peers and connecting to professionals were relevant or correct. The programs on perinatal-specific issues (Studies 1 and 6) also received high scores for content appropriateness and fitting users’ health status. Our finding was consistent with a previous finding that the perceived relevance of program content and personal circumstances were important for treatment engagement and adherence [25]. However, Study 2 focused on specific needs of working mothers with preschool children but did not receive high evaluations for appropriateness in terms of users’ health status or social condition. Study 2 was developed based on ACT theory [26], and contained metaphors and stories about people under high distress. Some users might have felt that the program was not appropriate for their health status or social condition. Researchers should understand the pros and cons of targeting participants’ specific issues and possibly consider a detailed assessment of those needs and adopt a segmentation strategy. In our study, appropriateness showed moderate or low associations with completion rate. However, belief in starting a program and treatment can be based upon perceived appropriateness, which indicates perceived efficacy in meeting needs and recognizing innovation for addressing personal issues and problems [27]. Improving appropriateness rather than adherence might benefit another side of the process.

### 4.3. Feasibility

For feasibility, we assessed the difficulty of using the program, time requirements for program participation, and content difficulty. Time seemed to be a key factor among programs with high feasibility evaluations. Programs that required less than 15 min to complete a session, with six sessions overall, received high evaluations. Study 5 received high feasibility evaluations because its movie sharing media (YouTube) took little time per session, which also shortened the duration of the whole course. Media sharing sites such as YouTube are potential resources for knowledge translation as they are easy to use and free of cost [28]. Furthermore, young participants in Study 5 might have found the YouTube intervention more feasible than a reading-based platform. In contrast, each session and the whole course of Study 6 took longer to complete, with users evaluating this program as less implementable. Simplifying content and reducing the number of sessions and content length might lead to higher program feasibility. As for completion rate, feasibility showed a moderate association with completion rate. Feasibility impacts the early phase of program adoption [8] because users’ motivation to continue using a system can reflect on the completion rate.

### 4.4. Satisfaction

Our study included satisfaction evaluations. Programs with high evaluations in other subcategories also showed high satisfaction scores. In implementation science, users’ perceived satisfaction with a program is important and future studies should explore the details. Regarding completion rate, satisfaction showed a high association with completion rate. Previous research suggests that high treatment satisfaction achieves high adherence [13]. Note that overall satisfaction with the iOSDMH is evaluated by only one item just once after the intervention, even though satisfaction has multifaceted or multilevel time-varying components [14]. For instance, program completion might promote high satisfaction but it remains unclear in this study.

### 4.5. Total Scores of the iOSDMH

The iOSDMH total scores reflected overall subscale evaluations, with Study 5 receiving the highest scores of all. Study 5 successfully reached the population in need, had appropriate programs utilizing a feasible method, and thus provided a sense of satisfaction. The iOSDMH total scores showed a positive correlation with program completion rate, suggesting that perceived evaluation of implementation aspects has some influence on completion rate. Although completion rate might be influenced more by reminder frequency, internal learning needs, and informal pressures for contextual use, implementation outcomes measured by the iOSDMH partially predict adherence. Revising the program to improve its total score might contribute to program effectiveness and efficacy through increased adherence. Because the programs were based on different psychological models and have different primary outcomes, total scores among different trials should be interpreted with caution. Due to differences among the trials, cutoff points for total scores were not reported in this study. The iOSDMH total scores might be beneficial especially when compared with studies having similar research designs and primary outcomes or to reevaluate implementation aspects of the program after modifying program contents or intervention delivery methods.

### 4.6. Implications of the iOSDMH

We found that iOSDMH subscales provided rich information on users’ evaluation of a program in terms of acceptability, appropriateness, feasibility, and overall satisfaction, and these measures seemed to be associated with program completion rate. Researchers can review each subscale score to fulfill unmet needs of the program contents or intervention design as perceived by users. Because this study discussed important factors for each item of implementation, researchers can benefit from our findings to find clues for improvement. After refining a program or study design, researchers can then readminister iOSDMH scales for reevaluation. Moreover, even when an intervention program fails to show a significant effect for primary endpoint outcomes, implementation outcomes would provide researchers essential information to explore the reasons an intervention program did not perform as expected, leading to future improvement of the program. In the clinical practice, using this scale provide the indicator of improvement progress. Comprehensive assessment of iOSDMH may enlighten the target area which needs to be improved and lead further refinement for contents or delivery-related strategies of digital mental health program in clinical settings.

### 4.7. Limitation

This study had several limitations. First, a cut-off point was not reported for each iOSDMH subscale or total score because each study had different study settings and objectives. Second, iOSDMH scales relied on users’ reported outcomes. Therefore, we attempted to compare iOSDMH scores with the program completion rate. Third, as the trials examined in this study consisted of a convenience sample of RCTs, our findings must be validated in other trials. Forth, the iOSDMH was not validated compared to other scales for measuring implementation outcomes of digital interventions. There is still room to examine further priority of the scale compared to other existing scales.

## 5. Conclusions

This study showed the effectiveness of the iOSDMH scale to evaluate essential aspects of digital mental health programs and serve as a significant indicator of program completion. Evaluation of implementation outcomes might also be important for maximizing effectiveness, which can be highly affected by completion rate. The iOSDMH scale can direct researchers toward future program goals to achieve social implementation and maximum effectiveness.

## Figures and Tables

**Figure 1 ijerph-19-15792-f001:**
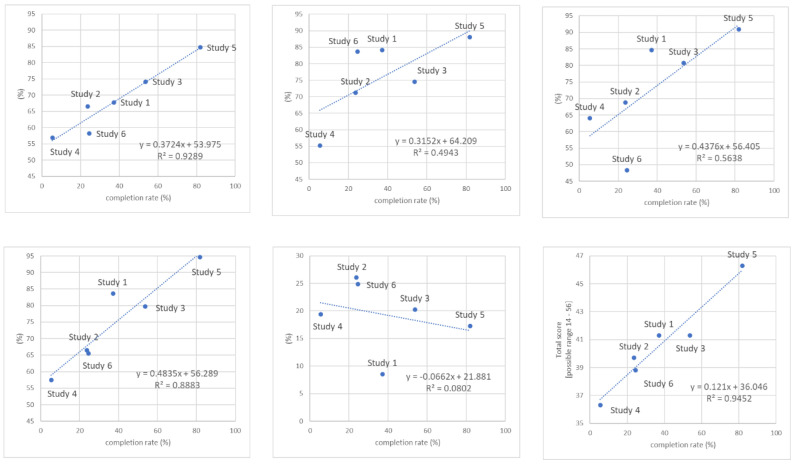
The association of implementation outcomes measured by iOSDMH with completion rate of the digital mental health programs.

**Table 1 ijerph-19-15792-t001:** Study Characteristics of Six Randomized Controlled Trials.

Variable	Study 1 iPDP	Study 2 Happiness Mom	Study 3 Smart CBT	Study 4 Imacoco-Care	Study 5 Online Student Peer GKT Program	Study 6 Smart Mama
Basal theory of intervention	Cognitive behavioral therapy	Acceptance and commitment therapy	Cognitive behavioral therapy	Multimodule consisting mainly of mindfulness, behavioral activation, and physical activity	Created with reference to the Ministry of Health, Labor, and Welfare guidelines and programs implemented by the nonprofit organization	Behavioral activation therapy
Number of sessions	6 modules	8 modules	7 modules (6 exercises in total)	6 modules	6 modules	12 modules with homework assignments
Learning time per session	5–10 min	6–30 min (average: 15 min)	5–10 min for a lecture; 5–20 min for an exercise	15 min	Average 14 min	15–30 min
Target population	Pregnant women	Working mothers with a preschool child	Full-time employees	Full-time employees	Vocational school students, junior college students, college students, graduate students, and college of technology students aged 18–29 in Japan	Postnatal women
Recruitment method	Pregnant women with user IDs for the app were sent an invitation message	Participants recruited from private companies and individuals through Facebook ads	Participants recruited from a pool of 300,000 people living in all 47 prefectures of Japan who registered with an online survey company	Recruited from registered members of a web survey company in Japan	Applied individually through publicity from collaborators by Snowbowl sampling, promotion through Twitter, Instagram, and websites	Recruited at postnatal hospital checkup (1 month postpartum)
Primary aim of intervention	Prevention	Prevention (to improve well-being)	Prevention	Prevention	Improve peer capabilities to prevent suicide	Prevention of depressive symptoms and abusive behaviors to children
Primary outcome	Onset of major depressive episode assessed by CIDI	Ryff’s psychological well-being at 6 months	Depression (BDI-II)	Psychological distress (K6) and Fear of the Coronavirus disease 2019 (COVID-19) infection (The Fear of COVID-19 Scale)	Gatekeeper Self-Efficacy Scale (GKSES)	EPDS and Conflict Tactics Scales-1 at 12 weeks (co-primary outcomes)
Total number of study participants (intervention group)	*N* = 5017 (*n* = 2509)	*N* = 841 (*n* = 424)	*N* = 1296 (*n* = 648)	*N* = 1200 (*n* = 600)	*N* = 321 (*n* = 160)	*N* = 124 (*n* = 62)
Intervention type	Smartphone-based e-learning	Web based e-learning	Web-based e-learning	Web-based psychoeducation	Web based e-learning	Web-based psychoeducation
Duration of intervention	12–16 weeks	12 weeks	10 weeks	1 month	10 days	12 weeks
Program completion rate	37.2%	23.7%	53.7%	5.5%	81.9%	24.6%
Content type	Multimedia self-help e-learning, mixed with text readings and voice guides	Multimedia self-help e-learning, mixed with text readings, voice guides, and writing exercises	Self-help e-learning with text readings, exercises guided by AI algorithm and chat-bots	Contents were mainly text, illustrations, video, and audio narration	Self-help e-learning with videos and a comment section (online discussion board)	Self-help e-learning with text readings, writing exercise, and mood diary assignment with feedback from therapists.
Homework/exercises	None	None	Yes	None	None	Mood diary assignments with feedback from therapists.
Availability	Can be accessed at any time they like	Can be accessed at any time they like	Can be accessed at any time they like	Can be accessed at any time they like	Can be accessed at any time they like	Can be accessed at any time they like
Interactions with professionals	None	None, but participants can receive comments from professionals upon request or comments on the communication board	None	None	None	Participants were offered opportunities to ask questions and receive comments from trained therapists
Interactions with other participants	None	Online sharing board where intervention group participants can read and write their thoughts and questions about the module (the researcher replied to comments)	None	None	Online discussion board where intervention group participants can read and write their thoughts about the module (no-reply system)	None
Timing of new module reminder	Participants were notified when a new module started	Email notification of new module start date (once a week for 8 weeks) followed by two additional contents and information	E-mail notification when a new module starts	During the intervention period (1 month), participants received two reminder e-mails	During the program period, we sent three emails informing participants of the days remaining until the end of the period	Participants received weekly reminders to promote learning during the intervention period
Additional reminders for non-learners	Intervention group participants received a popup message reminder to complete the program if they had not done so within a week after notification	Individual study progress reminder to promote learning (twice during the intervention period; 3rd and 6th weeks)	E-mail reminder of uncompleted modules (once a week for first 6 weeks). E-mail reminder of uncompleted modules with encouragement of repeated access to the program (once a week during 7th–10th weeks)	None	Participants who did not take the program within the program period received two reminders in a week after the program ended	None
Reward for participation	None	None	Intervention group participants received a token equivalent to JPY 1000 (USD 9.1) for completing all program modules. Control group participants received a token equivalent to JPY 100 (USD 0.91).	None	None	Intervention group participants received JPY 1000–3000 for both participating and completing surveys
Reward for completing the questionnaires	JPY 500 for each completed survey	None	JPY 30 (USD 0.27) for each completed survey (baseline, and 3-, 6-, and 12-month follow-up surveys)	Small token (data unavailable due to confidential information of the online panel company) for each completed survey	Among participants who answered all three surveys, 128 people had a chance to win a JPY 1200 Amazon gift card as monetary incentive to promote retention and follow-up completion by lottery	Control group participants received JPY1000–3000 for completing surveys
Respondents’ characteristics for the iOSDMH	Intervention group participants (*n* = 946)	Intervention group participants who learned at least one program module (*n* = 142)	Intervention group participants who learned at least one program module (*n* = 474)	Intervention group participants who viewed Imacoco Care at least once (*n* = 235)	Intervention group participants who learned at least one program module (*n* = 131)	Intervention group participants who learned at least one program module (*n* = 29)
Timing of the iOSDMH measurement	34 weeks gestation (12–16 weeks after enrollment)	12 weeks after enrollment	12 weeks after enrollment	4 weeks after enrollment	10 days after starting the intervention	12 weeks after enrollment

Note: iPDP: internet-based cognitive behavioral therapy for prevention of depression during pregnancy and in the postpartum period; GKT: gatekeeper training; CIDI: Composite International Diagnostic Interview; BDI-II: Beck Depression Inventory-Second Edition; EPDS: Edinburgh Postnatal Depression Scale.

**Table 2 ijerph-19-15792-t002:** iOSDMH Descriptive Scores.

			Disagree	Relatively Disagree	Relatively Agree	Agree	Preferable Responses and Harms	Group Difference (χ^2^ Test)
Item question (short item description)	Study	Total analytic sample	*n* (%)	*n* (%)	*n* (%)	*n* (%)	%	*p* value
**Acceptability**								
The advantages of my using this program outweigh the disadvantages for keeping my healthy mental health. (Advantages outweigh the disadvantages for keeping my mental health)	1	946	48 (5.1)	158 (16.7)	658 (69.6)	82 (8.7)	78.2	<0.001
	2	142	10 (7.0)	18 (12.6)	73 (51.0)	41 (28.7)	79.7	
	3	474	11 (2.3)	81 (17.1)	283 (59.7)	99 (20.9)	80.6	
	4	235	11 (4.7)	66 (28.1)	147 (62.6)	11 (4.7)	67.2	
	5	131	9 (6.9)	3 (2.3)	57 (43.5)	62 (47.3)	90.8	
	6	29	2 (6.9)	5 (17.2)	15 (51.7)	7 (24.1)	75.8	
Using this program improves my social image. (Improves my social image)	1	946	240 (25.4)	340 (35.9)	340 (35.9)	26 (2.7)	38.7	<0.001
	2	142	12 (8.4)	56 (39.2)	59 (41.3)	15 (10.5)	51.8	
	3	474	26 (5.5)	141 (29.7)	259 (54.6)	48 (10.1)	64.7	
	4	235	26 (11.1)	104 (44.3)	98 (41.7)	7 (3.0)	44.7	
	5	131	10 (7.6)	30 (22.9)	63 (48.1)	28 (21.4)	69.5	
	6	29	7 (24.1)	13 (44.8)	7 (24.1)	2 (6.9)	30.0	
This program is acceptable for me. (Acceptable for me)	1	946	16 (1.7)	113 (11.9)	664 (70.2)	153 (16.2)	86.4	<0.001
	2	142	10 (7.0)	35 (24.5)	69 (48.3)	28 (19.6)	67.9	
	3	474	15 (3.2)	93 (19.6)	259 (54.6)	107 (22.6)	77.2	
	4	235	14 (6.0)	83 (35.3)	124 (52.8)	14 (6.0)	58.7	
	5	131	3 (2.3)	5 (3.8)	59 (45)	64 (48.9)	93.9	
	6	29	3 (10.3)	6 (20.7)	11 (37.9)	9 (31.0)	68.9	
**Appropriateness**								
The content of the program is appropriate (from your perspective, it is the right thing to do). (Appropriate [from your perspective, it is the right thing to do])	1	946	6 (0.6)	75 (7.9)	699 (73.9)	166 (17.5)	91.4	<0.001
	2	142	7 (4.9)	24 (16.8)	75 (52.4)	36 (25.2)	77.6	
	3	474	14 (3)	59 (12.4)	292 (61.6)	109 (23)	84.6	
	4	235	12 (5.1)	59 (25.1)	150 (63.8)	14 (6.0)	69.8	
	5	131	2 (1.5)	3 (2.3)	64 (48.9)	62 (47.3)	96.2	
	6	29	0 (0)	2 (6.9)	17 (58.6)	10 (34.5)	93.1	
This program is applicable with my health status (e.g., pregnancy, physical and mental condition, etc). (Applicable to my health status)	1	946	20 (2.1)	111 (11.7)	659 (69.7)	156 (16.5)	86.2	<0.001
	2	142	11 (7.7)	33 (23.1)	68 (47.6)	30 (21.0)	68.6	
	3	474	21 (4.4)	121 (25.5)	277 (58.4)	55 (11.6)	70.0	
	4	235	17 (7.2)	100 (42.6)	111 (47.2)	7 (3.0)	50.2	
	5	131	6 (4.6)	16 (12.2)	66 (50.4)	43 (32.8)	83.2	
	6	29	1 (3.5)	4 (10.3)	19 (65.5)	6 (20.7)	86.2	
This program is suitable for my social conditions (e.g., work, housekeeping, commute, etc). (Suitable for my social conditions)	1	946	31 (3.3)	145 (15.3)	651 (68.8)	119 (12.6)	81.4	<0.001
	2	142	9 (6.3)	34 (23.8)	69 (48.3)	30 (21.0)	69.3	
	3	474	21 (4.4)	111 (23.4)	284 (59.9)	58 (12.2)	72.1	
	4	235	17 (7.2)	96 (40.9)	110 (46.8)	12 (5.1)	51.9	
	5	131	5 (3.8)	10 (7.6)	61 (46.6)	55 (42)	88.6	
	6	29	1 (3.5)	6 (20.7)	16 (55.2)	6 (20.7)	75.9	
This program fits with my living condition (e.g., place of residence). (Fits my living condition)	1	946	35 (3.7)	174 (18.4)	629 (66.5)	108 (11.4)	77.9	<0.001
	2	142	10 (7.0)	33 (23.1)	66 (46.2)	33 (23.1)	69.3	
	3	474	24 (5.1)	112 (23.6)	284 (59.9)	54 (11.4)	71.3	
	4	235	15 (6.4)	105 (44.7)	109 (46.4)	6 (2.6)	48.9	
	5	131	4 (3.1)	17 (13)	60 (45.8)	50 (38.2)	84.0	
	6	29	1 (3.5)	5 (17.2)	15 (51.7)	8 (27.6)	79.3	
**Feasibility**								
I believe this program is easy to use. (Easy to use)	1	946	23 (2.4)	157 (16.6)	594 (62.8)	172 (18.2)	81.0	<0.001
	2	142	11 (7.7)	45 (31.5)	65 (45.5)	21 (14.7)	70.2	
	3	474	17 (3.6)	91 (19.2)	265 (55.9)	101 (21.3)	77.2	
	4	235	8 (3.4)	92 (39.1)	123 (52.3)	12 (5.1)	57.4	
	5	131	3 (2.3)	13 (9.9)	55 (42)	60 (45.8)	87.8	
	6	29	2 (6.9)	10 (34.5)	13 (44.8)	4 (13.8)	58.6	
My using this program requires me physical effort (e.g., tired eyes, shoulder stiffness). (Physical effort) reverse item	1	946	459 (48.5)	386 (40.8)	92 (9.7)	9 (1.0)	10.7	<0.001
	2	142	39 (27.3)	56 (39.2)	44 (30.8)	3 (2.1)	32.9	
	3	474	154 (32.5)	208 (43.9)	88 (18.6)	24 (5.1)	23.7	
	4	235	47 (20.0)	127 (54.0)	57 (24.3)	4 (1.7)	26.0	
	5	131	57 (43.5)	56 (42.7)	12 (9.2)	6 (4.6)	13.8	
	6	29	5 (17.2)	14 (48.3)	8 (27.6)	2 (6.9)	34.5	
The total length of the program is implementable. (Total length is implementable)	1	946	16 (1.7)	159 (16.8)	607 (64.2)	164 (17.3)	81.5	<0.001
	2	142	11 (7.7)	38 (26.6)	71 (49.7)	22 (15.4)	65.1	
	3	474	14 (3)	101 (21.3)	244 (51.5)	115 (24.3)	75.8	
	4	235	13 (5.5)	87 (37.0)	128 (54.5)	7 (3.0)	57.4	
	5	131	5 (3.8)	8 (6.1)	60 (45.8)	58 (44.3)	90.1	
	6	29	3 (10.3)	14 (48.3)	9 (31.0)	3 (10.3)	41.3	
The length of 1 content is implementable. (Length of one content is implementable)	1	946	14 (1.5)	144 (15.2)	594 (62.8)	194 (20.5)	83.3	<0.001
	2	142	7 (4.9)	49 (34.3)	68 (47.6)	18 (12.6)	60.2	
	3	474	8 (1.7)	68 (14.3)	269 (56.8)	129 (27.2)	84.0	
	4	235	12 (5.1)	65 (27.7)	150 (63.8)	8 (3.4)	67.2	
	5	131	3 (2.3)	9 (6.9)	48 (36.6)	71 (54.2)	90.8	
	6	29	1 (3.5)	14 (48.3)	8 (27.6)	6 (20.7)	48.3	
The frequency of providing program is implementable. (Frequency is implementable)	1	946	6 (0.6)	59 (6.2)	618 (65.3)	263 (27.8)	93.1	<0.001
	2	142	2 (1.4)	40 (28.0)	73 (51.0)	27 (18.9)	69.9	
	3	474	4 (0.8)	60 (12.7)	286 (60.3)	124 (26.2)	86.5	
	4	235	9 (3.8)	80 (34.0)	141 (60.0)	5 (2.1)	62.1	
	5	131	2 (1.5)	6 (4.6)	58 (44.3)	65 (49.6)	93.9	
	6	29	1 (3.5)	9 (31.0)	13 (44.8)	6 (20.7)	65.5	
The program is easy to understand. (Easy to understand)	1	946	22 (2.3)	170 (18.0)	581 (61.4)	173 (18.3)	79.7	<0.001
	2	142	4 (2.8)	22 (15.4)	78 (54.5)	38 (26.6)	81.1	
	3	474	11 (2.3)	61 (12.9)	278 (58.6)	124 (26.2)	84.8	
	4	235	11 (4.7)	68 (28.9)	142 (60.4)	14 (6.0)	66.4	
	5	131	2 (1.5)	2 (1.5)	46 (35.1)	81 (61.8)	96.9	
	6	29	3 (10.3)	14 (48.3)	8 (27.6)	4 (13.8)	41.4	
**Overall Satisfaction**								
Overall, I am satisfied with the program.	1	946	21 (2.2)	137 (14.5)	636 (67.2)	152 (16.1)	83.3	<0.001
	2	142	11 (7.7)	36 (25.2)	58 (40.6)	37 (25.9)	66.5	
	3	474	13 (2.7)	83 (17.5)	283 (59.7)	95 (20)	79.7	
	4	235	11 (4.7)	89 (37.9)	124 (52.8)	11 (4.7)	57.4	
	5	131	3 (2.3)	4 (3.1)	56 (42.7)	68 (51.9)	94.6	
	6	29	2 (6.9)	8 (27.6)	13 (44.8)	6 (20.7)	65.5	
**Harms**								
Using this program causes physical symptoms (e.g., Tired eyes, headache, stiffness shoulders) (Physical symptoms)	1	946	486 (51.4)	340 (35.9)	110 (11.6)	10 (1.1)	12.7	0.026
	2	142	62 (43.4)	55 (38.5)	25 (17.5)	0 (0)	17.5	
	3	474	197 (41.6)	188 (39.7)	72 (15.2)	17 (3.6)	18.8	
	4	235	78 (33.2)	115 (48.9)	41 (17.4)	1 (0.4)	17.9	
	5	131	69 (52.7)	41 (31.3)	15 (11.5)	6 (4.6)	16.1	
	6	29	13 (44.8)	9 (31.0)	6 (20.7)	1 (3.5)	24.2	
Using this program causes mental symptom (e.g., depression, insomnia). (Mental symptoms)	1	946	605 (64.0)	291 (30.8)	44 (4.7)	6 (0.6)	5.3	<0.001
	2	142	78 (54.5)	50 (35.0)	11 (7.7)	3 (2.1)	9.8	
	3	474	237 (50)	170 (35.9)	56 (11.8)	11 (2.3)	14.1	
	4	235	90 (38.3)	111 (47.2)	33 (14.0)	1 (0.4)	14.5	
	5	131	74 (56.5)	32 (24.4)	19 (14.5)	6 (4.6)	19.1	
	6	29	13 (44.8)	10 (34.5)	6 (20.7)	0 (0)	20.7	
Using this program sometimes brings us a smart phone induced dangerous experience regarding safety (e.g., collide with people while walking and looking at the smart phone). (Induced dangerous experience regarding safety)	1	946	687 (72.6)	207 (21.9)	43 (4.5)	9 (1.0)	5.5	<0.001
	2	142	97 (67.8)	41 (28.7)	4 (2.8)	0 (0)	2.8	
	3	474	245 (51.7)	147 (31)	70 (14.8)	12 (2.5)	17.3	
	4	235	92 (39.1)	106 (45.1)	29 (12.3)	8 (3.4)	15.7	
	5	131	88 (67.2)	24 (18.3)	13 (9.9)	6 (4.6)	14.5	
	6	29	20 (69.0)	9 (31.0)	0 (0)	0 (0)	0	
I have a concern that the use of this program consumes my time for other activities (e.g., time for leisure, family affairs, sleep, education) (Time-consuming)	1	946	559 (59.1)	283 (29.9)	93 (9.8)	11 (1.2)	11.0	<0.001
	2	142	32 (22.4)	42 (29.4)	55 (38.5)	13 (9.1)	47.6	
	3	474	175 (36.9)	183 (38.6)	96 (20.3)	20 (4.2)	24.5	
	4	235	64 (27.2)	108 (46.0)	59 (25.1)	4 (1.7)	26.8	
	5	131	62 (47.3)	35 (26.7)	25 (19.1)	9 (6.9)	26.0	
	6	29	12 (41.4)	6 (20.7)	8 (27.6)	3 (10.3)	37.9	
Using this program makes me face the excessive pressure on learning this program regularly. (Excessive pressure on learning regularly)	1	946	593 (62.7)	278 (29.4)	69 (7.3)	6 (0.6)	7.9	<0.001
	2	142	33 (23.1)	34 (23.8)	64 (44.8)	11 (7.7)	52.5	
	3	474	173 (36.5)	176 (37.1)	107 (22.6)	18 (3.8)	26.4	
	4	235	78 (33.2)	105 (44.7)	46 (19.6)	6 (2.6)	22.1	
	5	131	79 (60.3)	38 (29)	12 (9.2)	2 (1.5)	10.7	
	6	29	9 (31.0)	8 (27.6)	8 (27.6)	4 (13.8)	41.4	

**Table 3 ijerph-19-15792-t003:** iOSDMH Scores and Subscales.

iOSDMH Subscales (Number of Items; Possible Range)	Study 1 (*n* = 934)	Study 2 (*n* = 142)	Study 3 (*n* = 474)	Study 4 (*n* = 235)	Study 5 (*n* = 131)	Study 6 (*n* = 29)	
	Mean (SD)	Mean (SD)	Mean (SD)	Mean (SD)	Mean (SD)	Mean (SD)	Group Difference (ANOVA)
Total (14 items; 14–56)	41.3 (5.7)	39.7 (8.7)	41.3 (6.9)	36.3 (6.5)	46.3 (6.6)	38.8 (7.4)	*F* = 44.7, *p* < 0.001
Acceptability (3 items; 3–12)	8.0 (1.5)	8.4 (2.2)	8.7 (1.7)	7.6 (1.7)	9.6 (1.8)	8.0 (2.1)	*F* = 32.8, *p* < 0.001
Appropriateness (4 items; 4–16)	11.9 (2.0)	11.5 (3.0)	11.4 (2.3)	10.1 (2.3)	13.0 (2.4)	12.3 (2.4)	*F* = 33.8, *p* < 0.001
Feasibility (6 items; 6–24)	18.5 (2.8)	17.0 (3.6)	18.3 (3.2)	16.0 (2.7)	20.3 (3.0)	15.8 (3.6)	*F* = 49.9, *p* < 0.001
Harm (5 items; 5–20)	7.4 (2.7)	9.4 (2.7)	9.0 (3.3)	9.3 (3.0)	8.2 (3.1)	9.2 (3.2)	*F* = 35.2, *p* < 0.001
Satisfaction (1 item; 1–4)	3.0 (0.6)	2.9 (0.9)	3.0 (0.7)	2.6 (0.7)	3.4 (0.7)	2.8 (0.9)	*F* = 29.4, *p* < 0.001

## Data Availability

The data that support the findings of this study (related to iOSDMH only) are available from the corresponding author, DN, upon reasonable request.

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
