# Peer review of "Usefulness of Implementation Outcome Scales for Digital Mental Health (iOSDMH): Experiences from Six Randomized Controlled Trials"

_ijerph, 2022, doi:10.3390/ijerph192315792_

Round 1
Reviewer 1 Report
I read the paper "Usefulness of Implementation Outcome Scales for Digital Mental Health (iOSDMH): Experiences from six randomized controlled trials" with great interest.
The instrument that the authors developed and, in this paper, tested across six different randomised controlled studies can be an extremely useful tool for future studies that aim to examine implementation outcomes or conduct process evaluations along with Trials.
The paper has a very good structure and the methodology, and the presentation of the findings are scientifically sound. My recommendation is the authors to add some information either in their introduction or conclusion about other current scales or methods that exist and measure separately any of the factors that their scale measures and discuss what they think may be the advantages for using the iOSDMH scales. Also, if during the development of the scales had already used it in any other scales in order to examine parameters such as their convergent validity it would also be helpful to briefly state any of their findings.
Overall, though, it is a highly interesting and, in my opinion, important study.
Author Response
We attached the point-by-point response to the reviewer’s comments.

Reviewer 2 Report
Thank you for the opportunity to review this interesting and well-written paper. Overall, the information is presented extremely well, however there are some minor amendments which would improve this paper:
Firstly, it would be useful to know how this differs from other well-established implementation frameworks like the Proctor taxonomy (you mentioned that you based the iOSDMH on this in the introduction, but it is important to know why this additional tool was needed). Perhaps this can be expanded on in the introduction, in addition to reference to other implementation frameworks.
I recommend cutting down the text on page 12 - I found this quite difficult to follow and repetitive.
Currently the implications section seems quite thin - are there any implications for clinical practice (e.g. use of interventions in clinical setting, rather than research), and why and in which situations should researchers use this tool rather than other established implementation frameworks. Whilst this paper was an interesting read, the argument around the utility of this tool needs to be made more explicit. Otherwise, I am unsure of why I should use this measure above any alternatives - what gap does this fill?
Author Response
We attached the point-by-point response to the reviewer's comments.
